# High-Speed and High-Temperature Calorimetric Solid-State Thermal Mass Flow Sensor for Aerospace Application: A Sensitivity Analysis

**DOI:** 10.3390/s22093484

**Published:** 2022-05-03

**Authors:** Lucas Ribeiro, Osamu Saotome, Roberto d’Amore, Roana de Oliveira Hansen

**Affiliations:** 1Instituto Tecnológico de Aeronáutica, Pça. Mal. Eduardo Gomes 50, São José dos Campos 12228-900, Brazil; osaotome@gmail.com (O.S.); damore@ita.br (R.d.); 2NanoSYD Centre, Mads Clausen Institute, University of Southern Denmark—SDU, Alsion 2, 6400 Sønderborg, Denmark; roana@mci.sdu.dk

**Keywords:** TMFS, flow sensor, calorimetric, MEMS, Air Data System, sensitivity

## Abstract

A high-speed and high-temperature calorimetric solid-state thermal mass flow sensor (TMFS) design was proposed and its sensitivity to temperature and airflow speed were numerically assessed. The sensor operates at 573.15 Kelvin (300 °C), measuring speeds up to 265 m/s, and is customized to be a transducer for an aircraft Air Data System (ADS). The aim was to enhance the system reliability against ice accretion on pitot tubes’ pressure intakes, which causes the system to be inoperative and the aircraft to lose protections that ensure its safe operation. In this paper, the authors assess how the distance between heater and thermal sensors affects the overall TMFS sensitivity and how it can benefit from the inclusion of a thermal barrier between these elements. The results show that, by increasing the distance between the heater and temperature sensors from 0.1 to 0.6 mm, the sensitivity to temperature variation is improved by up to 80%, and that to airspeed variation is improved by up to 100%. In addition, adding a thermal barrier made of Parylene-N improves it even further, by nearly 6 times, for both temperature and air speed variations.

## 1. Introduction

In an aircraft, critical flight data are provided to the cockpit by the Air Data System (ADS). The consolidated technology used to transduce the external airspeed measurements into digital information for other aircraft systems is the pitot tube, which performs this task by reading the dynamic and static intake air pressures at flight altitude [1,2].

Although the ADS is rigorously designed and qualified to operate in harsh environments, incidents or even accidents may occur that are related to the freezing of the pitot tubes’ orifice, despite its heating capabilities [3]. Once ice crystals accumulate in the transducer, the data become noisy (or lost), leading the air data computer software to ignore the defective probe. In recent decades, several incidents associated with this failure event have been reported [4,5,6]. To address this scenario, the Federal Aviation Administration (FAA) issued Advisory Circular (AC) 91-74B to pilots, which provides instructions about how to identify the atmospheric conditions that lead to this particular risk and how to plan the flight route to avoid supercooled clouds [7]. These instructions impose operational restrictions on the airplane mission, such as changes to a scheduled route, thereby increasing the associated operational costs.

A calorimetric thermal mass flow sensor (TMFS) is designed to transduce the external aircraft airspeed information into electrical signals for an ADS system, and aims to add an extra measurement technique to improve the data voting and reliability when flying in atmospheric conditions that can lead to ice accretion on the pitot’s orifice. The technology related to TMFS has been explored for many aerodynamic applications. These mainly focus on the capability to detect and control the airflow pattern on airfoils with the intention to improve its performance by reducing the drag caused by turbulent flows [8,9,10,11,12,13]. Sturm et al. and Leu et al. present TMFS designs that aim to detect the airflow boundary layer separation from the airfoil while pitching the surface to better control its angle of attack. For the application proposed herein, boundary layer detachment must be avoided so the speed readings do not become noisy. Therefore, the sensor is ideally installed in an articulated airfoil that can ensure the airflow is parallel to the TMFS regardless of the aircraft’s maneuvers.

A review of the literature shows that the maximum measured airflow speed reported by similar TMFS designs is limited to up to 100 m/s, which is much lower than the level necessary for the proposed application, which may be as high as 265 m/s [14,15,16,17,18,19,20,21,22,23]. Additionally, most of these designs have a heater built into a suspended membrane over a cavity to better manage the conductive heat in the sensor body, which is not recommended for the intended application due to high speeds, vibrations, and other mechanical stress factors that may lead to the collapse of the membrane. TMFS designs exist that do not rely on this technique to build the heater element; examples include the design of Glatzl et al., which operates at low temperatures to avoid thermal conductivity issues that can collapse the sensor, or those of Kaltsas et al., Petropoulos et al., Kim et al., Mayer et al., and Bruschi and Piotto, which make use of flow channels to ensure that the fluid flow is laminar, consequently reducing the power demand from the heater once the flow volume is controlled by the tube diameter. However, these are not suitable for the proposed application due to the aforementioned ice condition [24,25,26,27,28,29,30,31].

The TMFS sensitivity is affected by many design constraints, such as the materials employed, the technology used to build the thermal sensors, the sensors’ intrinsic sensitivity to temperature variation, and how well the thermal sensor is isolated from the heater. Regarding TMFS geometry, Mehmood et al. presents a relatively recent review of over 60 different TMFS designs, assesses how the membrane shape affects the sensor sensitivity, and determines the best membrane to heater length ratio (MHR) to be 3.35 for square membranes and 3.30 for circular membranes. Kim et al. evaluates how the distance between heater and temperature sensors affects the sensitivity by increasing the distance in four steps of 4 mm for a sensor built inside a flow channel. The study conclusion is that the TMFS sensitivity worsens when exposed to nitrogen gas as the distance from the heater to the temperature sensor increases [23,27].

In this study, the proposed TMFS calorimetric sensitivity was evaluated by incrementally varying the distance between the heater and the temperature sensors, and by including a thermal barrier made of Parylene-N to better isolate the heater and temperature sensors. The insulator was chosen due to its low thermal conductivity and compatibility with the microfabrication process. In this process, filling of the insulation trench in the SiC substrate can be achieved by the Chemical Vapor Deposition (CVD) process, and the excess is further etched using the Reactive Ion Etching (RIE) process via oxygen gas [28,29]. No change in the heater geometry was made or assessed. The assessment was accomplished through COMSOL Multiphysics and Simulink high fidelity simulations. The former is a Computational Fluid Dynamic (CFD) simulation, which is static and represents a single point of operation, and the second is a dynamic simulation covering an assumed flight envelope, thus allowing the sensitivity to be calculated. The CFD results validate the dynamic simulation when the same boundary conditions apply. No tuning between the models was performed. The results presented herein define the TMFS configuration in which its sensitivity benefits most from the assessed configurations.

The presented calorimetric TMFS design is innovative regarding the manner in which the conductive heat is managed. No similar designs making use of solid materials to isolate sensor elements have been proposed, and no existing designs operate in open circuits that do not rely on flow channels or tubes to take the flow measurements. In this sense, the proposed high-speed and high-temperature solid-state calorimetric TMFS will overcome a necessary gap to become a viable solution for an ADS application.

## 2. Proposed Calorimetric TMFS Application

The envisioned calorimetric TMFS installation must be located external to the aircraft and on the surface of an articulated airfoil similar to that used for angle of attack sensors. Thus, the non-heated start length is known, and the air flow is maintained to be as stable as possible and is independent of airplane maneuvers. Figure 1 presents an illustration of the calorimetric TMFS installation.

The calorimetric TMFS design is composed of a heater element with two temperature sensors at its opposite ends, so that the air flow crosses these elements perpendicularly. The temperature sensors were designed as Schottky diodes working as differential proportional to absolute temperature (PTAT) sensors. This type of diode has the advantage of a linear voltage–temperature dependence, long-term stability, low turn-on voltage, and faster recovery time when compared with the junction diode [32,33,34]. Additionally, by making use of silicon carbide (SiC) as the substrate to fabricate the Schottky diodes, it is possible to enhance the temperature sensor operation to up to 1000 °C [33,34,35,36,37].

With the objectives of increasing the TMFS sensitivity to heat convection and minimizing the thermal noise due to heat conduction through the TMFS substrate, the idealized calorimetric TMFS was assessed by incrementally increasing the distance between the heater and thermal sensors, and further adding a thermal insulation barrier. Figure 2 presents an illustration of the proposed sensor geometry.

The temperature sensors and the heating element are formed by gold and tungsten, respectively, to take advantage of the metals’ electrical properties. The selected substrate for the TMFS is SiC, which enables the PTAT Schottky temperature sensors to be fabricated.

## 3. Physics of the Application

A few basic laws are fundamental to the subject of this study: the law of conservation of mass, the law of conservation of momentum, and the law of conservation of energy. The interaction of the fluid flow with the surface over which the fluid flows creates the boundary layer, which has three horizontal flow patterns: laminar, transition, and turbulent; and three vertical flow patterns: viscous sublayer, buffer layer, and turbulent. Above the streamline, the fluid flow is free from any dynamic interference from the surface, and is called the free stream region. Despite its simplicity, the parallel flow over a flat plate occurs in numerous engineering applications, including this work; Figure 3 presents an image of how it develops.

The relative longitudinal position of the calorimetric TMFS heater on a flat plate determines the relation between the heat transferred by convection and the flow speed providing the dimensioning requirements for the heating element, i.e., the amount of power the heater has to generate to allow the speed measurement (heat transfer rate). The conductive heat from the calorimetric TMFS heater to temperature sensors through the substrate affects the overall sensitivity of the sensor, which should ideally measure only the convective heat.

### 3.1. Heat Convection

A measure of the flow behavior is given by the Reynolds Number, which defines the distance at which the flow becomes turbulent; at this point, more power is demanded to calculate the velocity once the heat convection with the air intensifies. The flow transition point is then derived from the Reynolds Number equation, which represents the ratio of the inertia to viscous forces and is described as [38]:(1)$$Re_{x}=\frac{\rho u_{\infty }x}{\mu },$$
where $u_{\infty }$ is the free stream velocity, x is the point over the flat plate where it is desired to obtain the Reynolds Number, ρ is the fluid density, and μ is the fluid viscosity. The transition from laminar to turbulent flow is obtained from Equation (1) based on the transition distance $x_{c}$, instead of x; this defines the critical Reynolds Number, which can vary from approximately $10^{5}$ to $3\times 10^{6}$, depending on the surface roughness and the turbulence level of the free stream.

To determine the amount of heat transfer needed to measure the desired speed, it is necessary to solve the boundary layer equations for the flat plate, starting with the development at the leading edge (x), and then obtaining the temperature profile, considering that there is a non-heated starting length (ξ) until the flow reaches the heating element (${q_{s}}^{\prime \prime }$). Figure 4 provides an illustration of this, where δ is the fluid boundary layer and $\delta_{t}$ is the thermal boundary layer.

With a known non-heated starting length (ξ), the Nusselt Number, a dimensionless parameter used to characterize the enhancement of the heat transfer due to convection, can be calculated. The Nusselt Number, $Nu_{L}$, is defined as the ratio of the heat transferred from a surface to the heat conducted away by the fluid being defined, for a flat plate with a constant average heat transfer, as [39]:(2)NuL¯=NuL¯ξ=0LL−ξ1−ξLp+1p+2pp+1,
where ξ is the distance of the non-heated part from the leading edge of the flat plate to the heating element; p is a constant that represents the flow pattern, which is equal to 2 for laminar flow and 8 for turbulent flow; and ${\left.\bar{Nu_{L}}\right|}_{\xi =0}$ is the average Nusselt Number for a plate of length L when heating starts at the leading edge, which is expressed by [39]:(3)NuL¯=0.680·ReL12·Pr13,
where Pr is known as the Prandtl Number, which is defined as the ratio of the kinematic viscosity to the thermal diffusivity and is, therefore, a fluid property.

Then, with Reynolds and Nusselt Numbers in place, it is possible to obtain the needed average heat transfer rate, in W/m^2^ K, to measure the airflow as [39]:(4)$$\bar{h_{L}}=\frac{\bar{Nu_{L}}\cdot \kappa }{L},$$
where κ is the fluid thermal conductivity.

The electrical power needed to be generated by the heated element to measure the intended flow speed is equal to the total heat transfer rate times the heating element area times the temperature difference between the heater and the fluid, and is expressed by [39]:(5)$$q_{conv}=\bar{h_{L}}\cdot A_{h}\cdot \left(T_{h}-T_{f}\right),$$
where $A_{h}$ is the heater surface area, $T_{h}$ is the heater temperature, and $T_{f}$ is the fluid temperature.

### 3.2. Heat Conduction

To examine the heat transfer by conduction, it is necessary to relate it to mechanical, thermal, or geometrical properties so the general form of the heat conduction equation can be expressed by the Fourier’s law [38]:(6)$$\frac{\partial^{2}T}{\partial x^{2}}+\frac{\partial^{2}T}{\partial y^{2}}+\frac{\partial^{2}T}{\partial z^{2}}+\frac{q}{k}=\frac{1}{\alpha }\frac{\partial T}{\partial t}$$
where k is the material thermal conductivity, q is the volumetric heat generation rate, and α=kρcp is known as the material thermal diffusivity composed of ρ, the material density and, $c_{p}$ the material specific heat capacity.

The solution for Fourier’s equation when the TMFS boundary conditions apply can be summarized as a unidimensional problem with the heat being conducted in the medium as a perpendicular vector flowing from the hottest to the cooler parts of the sensor. The implications of this approach are that Equation (6) is reduced to:(7)$$\alpha \frac{\partial^{2}T}{\partial x^{2}}+\frac{q}{\rho c_{p}}=0.$$

The temperature sensor measurements are then affected by the amount of heat being conducted through the TMFS body and the amount of heat being exchanged by convection. The upstream and downstream temperature sensors sense this phenomenon in different manners when the upstream sensor is measuring cooler air that has not been heated by the TMFS heater, which is not the case for the downstream sensor.

## 4. Materials and Methods

The results presented herein were obtained by numerical simulations. The COMSOL Multiphysics modeling enables CFD analysis by numerically resolving the thermal conductivity and convection equations based on the TMFS geometry when it interacts with air flowing on its top within fixed boundary conditions. The numerical solution provided by this software is highly reliable and represents, with confidence, the physical phenomena involved; however, it is computationally demanding. The high-fidelity Simulink model was created to dynamically observe the TMFS behavior when operating in the different configurations under evaluation, allowing the sensitivity to be calculated. In addition, the cost of the Simulink computational is much lower than that of the COMSOL Multiphysics simulation. The results obtained by both simulations were compared when operating under the same boundary conditions so to validate the Simulink model.

### 4.1. COMSOL Multiphysics

The designs presented in Figure 2a,b were modeled using the COMSOL Multiphysics tool, where the junction temperature of each thermal sensor metal was measured after the problem was numerically resolved by the software.

For comparison, 2D and 3D simulations were conducted and showed that the y-direction temperature dependence on the calorimetric TMFS body temperature distribution was limited to a small magnitude. This was because the low cross-section area available for the thermal conductive path formed in a way that could not significantly impact the results obtained from a 2D simulation. Therefore, each proposed design was modeled by creating bidimensional geometries, as presented in Figure 5, to save computational resources, and to allow a solution for heat transfer in a Solids and Fluids Multiphysics simulation to be computed. Additionally, this approach is highly used in CFD simulations because it reduces the size of the meshes generated without compromising the outcome results [40,41]. The TMFS element properties used during COMSOL Multiphysics simulations were as presented at Table 1 [42].

The CFD simulation represents a static solution for a specific moment in the flow pattern under analysis as a picture at an instant in time. This particular moment is represented by the air flow properties, which comprise the boundary conditions for the numeric solution to be computed, as shown in Table 2, with an air speed equal to 0.3 Mach (101.03 m/s) [43].

On top of the object under study, a two-dimensional volume was created that represents the mass of air traveling on the calorimetric TFMS, where the air intake and outtake were the left and right face of the volume, respectively. The non-heated starting length ξ was defined as 1 cm. The mesh was generated in the software by user-controlled options, where the maximum element size was 1.6 mm, the minimum element size was 0.1 μm, the maximum element growth rate was 1.05, the curvature factor was 0.2, and the resolution of narrow regions was equal to 1. The TMFS geometric dimensions were the same as presented in [44].

### 4.2. Simulink

The Simulink model was created by capturing all physical and dynamic characteristics of the atmospheric variation while altitude and speed were varied from 1000 to 10,000 m and 101.03 to 300 m/s, respectively, using the “ISA Atmospheric Model” library block. The International Standard Atmosphere (ISA) model, provided by the International Civil Aviation Organization (ICAO), supplies the simulation with air properties such as external temperature, thermal conductivity, specific heat, density, viscosity, and speed of sound at a specific altitude [43].

The calorimetric TMFS interaction with the airflow (convection) and the sensor body (conduction) were modeled using elements from the Simscape Simulink thermal library. Figure 6 depicts the block diagram representing the Simulink model. The airflow parameters are obtained from the “Atmospheric Data” block, which then feed the “Thermal Sensor” blocks, upstream and downstream, where the convective and conductive thermic data are calculated as presented in Section 3. Later, the metal junction temperatures are used to determine the Schottky diode variations, which in turns feed the “PTAT readout circuitry”, where “V_out_” is obtained. The “Heater” block implements the 132 Ω thermal resistance dimensioned using Equation (5), which generates the needed electrical power of 1.7 Watts to measure cruise speeds [44]. The dynamic behavior simulated by this model provides evidence about the TMFS sensor operation that supports the proposed air temperature and speed sensitivity analysis.

The speed and altitude inputs were modeled as being constant in the first 0.2 s so the atmospheric/boundary conditions were the same as those used in the COMSOL Multiphysics simulation; then, they were increased during the period from 0.2 to 2 s by varying the speed range up to 300 m/s and the altitude range up to 10,000 m, as previously described.

To better visualize how the Schottky diode junction temperature is simulated in the thermal sensor blocks in Figure 6, the conductive heat in the calorimetric TMFS body is represented by thermal resistances, as in Figure 7, for both assessed cases: without and with a thermal barrier. The model was divided into two layers: one layer representing the SiC wafer having a 0.3 mm thickness, which does not interact with the airflow (heat transfer on solids only), and another layer on top of the SiC wafer where the calorimetric TMFS elements are built in, having a 1 μm thickness, which interacts with the airflow (heat transfer on solids with convection to a fluid).

The Simscape block used to simulate the heat transfer on solids in the first layer of the SiC wafer is the conductive heat transfer, which represents the heat transfer by conduction between two layers of the same material, and is governed by the Fourier law. This remarkable law is used to derive a particular solution from the equations in Section 3.2. Figure 8 presents an image of the conductive heat transfer block and its associated parameters.

For the second layer of the calorimetric TMFS, the heat conduction problem has an amount of energy that is exchanged with the fluid by convection. Simulink Simscape does not have a block that models the conjugate heat transfer by conduction and convection with a fluid. Therefore, to determine the Schottky diode metal junction temperature, Equation (7) was resolved for the unidimensional heat conduction with heat transfer by convection to a fluid, and its solution is as follows:(8)$$T_{s}=\frac{T_{q}+\left(hd/k\right)T_{f}}{hd/k+1},$$
where $T_{s}$ is the Schottky diode metal junction temperature; $T_{q}$ is the conductive temperature from the SiC wafer; h is the heat transfer rate calculated using Equation (4), which varies with airflow speed (due to Reynolds Number influence) and the distance L from the flat plate leading edge; d is the Schottky metal thickness; k is the fluid thermal conductivity; and $T_{f}$ is the fluid temperature.

The differential PTAT circuit is depicted in Figure 9, where D_1_ and D_2_ are the upstream and downstream Schottky diodes, respectively, I_1_ and I_2_ are current references to polarize the diodes, V_in_ is equal to 12 Vdc, and A_1_ is an amplifier with gain equal to 12.

During the simulation, the heater was kept at a constant temperature of 573.15 Kelvin (300 °C). The Schottky diode Simulink model was created using the design proposed by Krasnov et al. [45], in which a 4H-SiC extreme temperature sensor is depicted. With the intention to validate the created Schottky diode model, the voltage–temperature dependence curves for electrical currents varying from 1 pA to 0.1 mA obtained from the Simulink model are presented in Figure 10, in addition to their respective sensitivities in mV/K.

All Simulink simulations, from the Schottky diode validation above to the TMFS dynamic behavior presented in Figure 6, were executed using a fixed-step of 1 μs and automatic solver selection.

### 4.3. Calorimetric TMFS Fabrication Process

The calorimetric TMFS thermal barrier can be fabricated by making use of standard cleanroom microfabrication techniques. The proposed process is only a coarse description of an actual fabrication process flow that is, in practice, much more complicated. This process can start with a commercial SiC wafer (300 µm thick). A photoresist is spin-coated onto the wafer, following a UV exposure using a photomask with the thermal barrier area design. The area for SiC etching is defined by the photolithography process, and a trench can be etched on SiC via Inductive Coupled Plasma RIE, as described in [45,46]. Then, a new photolithography process is conducted to define the areas for Parylene-N deposition, which can then be performed via CVD. The photoresist is then removed in an acetone bath, and the SiC wafer containing the Parylene-N trenches is cleaned via RCA cleaning.

The Schottky diode electrodes and heater can be defined by a subsequent photolithography process, followed by tungsten and gold deposition via e-beam metal deposition. Resist removal and lift-off are performed with an acetone bath. The final step is the dicing of the sensors with a diamond dicing saw and wiring-bonding to a PCB.

## 5. Results

Both COMSOL Multiphysics (static) and Simulink (dynamic) simulations were executed by incrementally increasing the distance between the heater and the sensing elements, i.e., distance “b” as defined in Figure 2a, by steps of 0.1 mm from the initial value of 0.1 mm to 0.6 mm. When reaching this value, the distance was held constant, but an insulation trench was added. This was filled with Parylene-N having an initial thickness, “IT_h_”, as defined in Figure 2b, of 0.2 mm; this was incrementally increased by 0.1 mm to 0.4 mm. The results obtained from these simulations are presented hereafter.

### 5.1. COMSOL Multiphysics

By incrementing the distance between the heater and the temperature sensors, and later adding the insulation trench made with Parylene-N, the mesh was automatically generated in the tool as stated in Section 4.1; the quantity of the nodes and the minimum element quality of the domain and boundary elements are presented in Table 3.

The delta temperature $\left(\Delta T\right)$ between the upstream and downstream Schottky diodes’ metals was measured and is presented in Figure 11, in Kelvin.

Figure 12 presents an image of the COMSOL Multiphysics solution computed for each condition assessed, where, from letter (a) to (f), the distance between the heater and thermal sensors is increased from 0.1 to 0.6 mm, and a small variation in the shade of red can be seen; from letter (g) to (i), the thickness of the thermal barrier is varied from 0.2 to 0.4 mm, and a shade of yellow starts to develop, showing a lower temperature on the thermal sensor region and indicating the heat conduction was partially blocked through the TMFS substrate.

### 5.2. Simulink

As mentioned previously, in the first 0.2 s of the Simulink simulation, the airflow boundary conditions were kept at the same values as those used in the COMSOL Multiphysics CFD simulation for validation and comparison purposes. After this time, speed and altitude were linearly increased to 300 m/s and 10,000 m respectively. The obtained ΔT in Kelvin for each condition is presented in Figure 13.

The ΔT results obtained from COMSOL Multiphysics and Simulink simulations are then compared in Table 4. It is possible to notice differences due to the solver configurations of both software packages, which led to numerical discrepancies, and due to the thermal inertia captured in the Simulink model for the cases where the thermal barrier was present.

To calculate the overall TMFS sensitivity to temperature variation from the Simulink model, V_out_ at the PTAT circuit was linearized using the “polyfit” and “polyval” functions for *t* > 0.2, for each condition assessed. The calorimetric TMFS sensitivity values to temperature variation are presented in Figure 14, where the linearized voltage curves and their respective slopes, or sensitivity, are highlighted. It is possible to notice the improvement as the distance between the heater and thermal sensors increases (foreground graphic), and then with the addition of the thermal barrier (background graphic).

To calculate the overall TMFS sensitivity to air speed variation from the Simulink model, V_out_ at the PTAT circuit was linearized using the “polyfit” and “polyval” functions for *t* > 0.2, for each condition assessed. The calorimetric TMFS sensitivity to air speed variation is presented in Figure 15, where the linearized voltage curves and their respective slopes, or sensitivity, are highlighted. It is possible to notice the improvement as the distance between the heater and thermal sensors increases (foreground graphic), and then with the addition of the thermal barrier (background graphic).

## 6. Discussion

The simulation results presented in the previous section show a consistent improvement in the sensitivity to both temperature and speed variations of the proposed high-speed and high-temperature calorimetric solid-state TMFS when varying the distance between the heater and thermal sensors, and then including Parylene-N as a thermal barrier between them.

The results obtained via COMSOL Multiphysics CFD simulation presented in Figure 12 show how the heat propagates through the sensor body. As a result, it is possible to visualize the improvements in the Schottky diode metal junction temperature, which benefits the temperature difference between the upstream and downstream thermal sensors, as presented in Figure 11.

Due to the computational costs associated with COMSOL Multiphysics CFD simulation, which resolves the heat propagation problem by iteratively solving the heat transport from one mesh node to another until the convergence criteria are established by the software, the Simulink model was created. The solutions from both software packages were further compared, as presented in Table 4, for the same boundary conditions. The results were found to be similar, with an acceptable error margin, when the thermal barrier was included, validating the Simulink results for this condition. For the condition where there was no thermal barrier, the results from Simulink showed a significant error for b = 0.1 mm and b = 0.2 mm. This was caused by a numerical error due to the short distance “D” on the conductive heat blocks, which appears to be a limitation of the Simscape block. The sensor sensitivity to air temperature and air speed variation was then calculated from dynamic results obtained in Simulink, and showed significant improvements in the overall sensor sensitivity, as presented in Figure 14 and Figure 15. Furthermore, the addition of Parylene-N as a thermal barrier was proven to be highly efficient at reducing the heat conduction through the TMFS body, thus resolving the issue reported in previous work [44].

Here, it is important to comment that the results presented by Kim et al. [27] show the opposite behavior, where the sensitivity decreases as the distance between the temperature sensors and the heater increases. This is explained by the fact that the authors used a flow channel, which requires lower power from the heater (0.8 Watts) than the application proposed herein (1.7 Watts), and nitrogen gas, which is lighter than air, to perform the tests that support this conclusion. The heat convection with the fluid dissipated the energy from the heater operating at 160 °C; as a result, the more distant temperature sensors were unaffected, thus decreasing the overall sensor sensitivity. The application proposed herein does not rely on flow channels, operates at 573.15 Kelvin (300 °C), and has different boundary conditions. Therefore, the results presented by Kim et al. [27] do not form a basis for comparison with the presented results. Furthermore, previous work has shown that the thermal plume generated by the heater while operating at cruise speeds is prominent. This allows the temperature sensors placed at the distances used in the current study to measure the convective effect resulting from the airflow interaction with the proposed sensor [44].

Based on the results presented herein, the final TMFS dimensions, using a thermal barrier of 4 mm filled with Parylene-N and a distance between the heater and thermal sensors of 6 mm, can be defined, thus allowing it to be manufactured and tested in wind tunnels for the intended operation.

## 7. Conclusions

From the COMSOL Multiphysics and Simulink simulation results, it is possible to observe that the delta temperature $\left(\Delta T\right)$ measured between thermal sensors increases as the distance between the heater and the thermal sensors increases for airflow speeds and altitude variations representing a generic aircraft flight envelope. By subsequently adding a thermal barrier made of Parylene-N, the ΔT measured between thermal sensors increased even further, showing it to provide good thermal insulation and fit for its purpose.

The results obtained from COMSOL Multiphysics validate the Simulink simulation when compared under the same boundary conditions, with the exception of b = 0.1 mm and b = 0.2 mm, for the aforementioned reasons. The ΔT between the temperature sensors increases as the air temperature decreases, showing that the sensor operation at cruise speeds can yield a better flow measurement due to the high thermal convection caused by the air flow dynamics. It is important to also note that the thermal barrier has a thermic delay. Further investigation will be required about how this behaves when operating during long and continuous periods of time to ensure the benefit observed is permanent.

Furthermore, the improvements observed in the ΔT measured between the thermal sensors for the different simulated conditions show that the calorimetric TMFS sensitivity to air temperature and air speed variations was significantly improved.

## Figures and Tables

**Figure 1 sensors-22-03484-f001:**
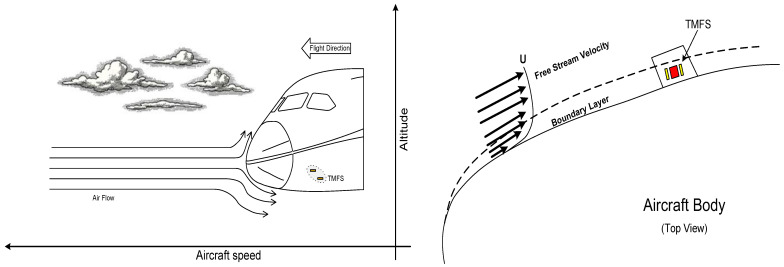
Idealized TMFS installation.

**Figure 2 sensors-22-03484-f002:**
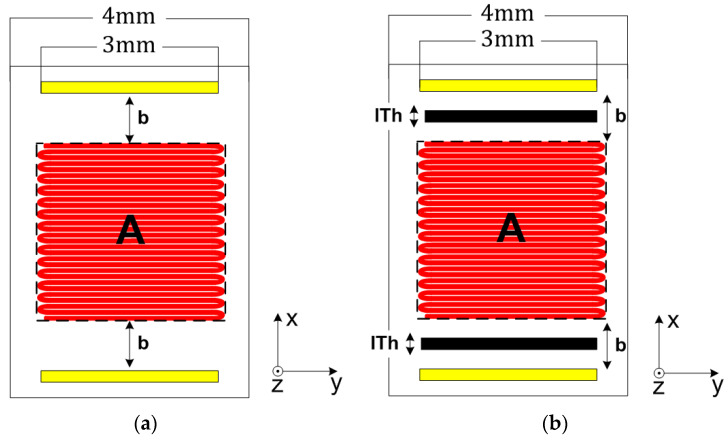
Top view of the idealized TMFS layout (**a**) without and (**b**) with thermal barrier, where the red square “A” represents the heated area, the yellow rectangles are the temperature sensor metal, “b” is the distance between the heater and temperature sensors, and “ITh” represents the insulation barrier thickness.

**Figure 3 sensors-22-03484-f003:**
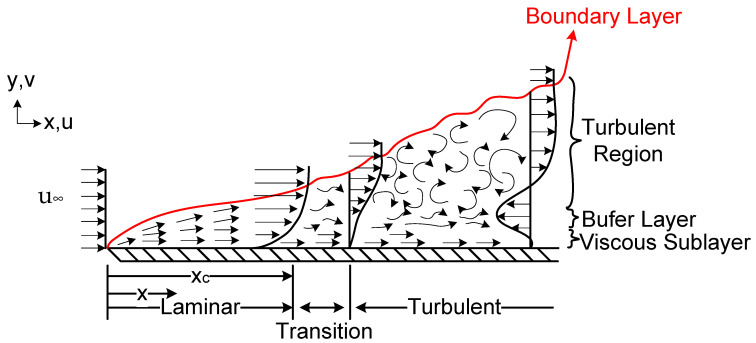
Velocity boundary layer development on a flat plate.

**Figure 4 sensors-22-03484-f004:**
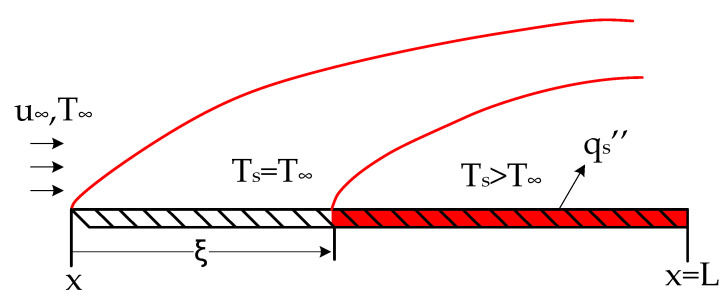
Thermal boundary condition for on a flat plate with unheated starting length.

**Figure 5 sensors-22-03484-f005:**
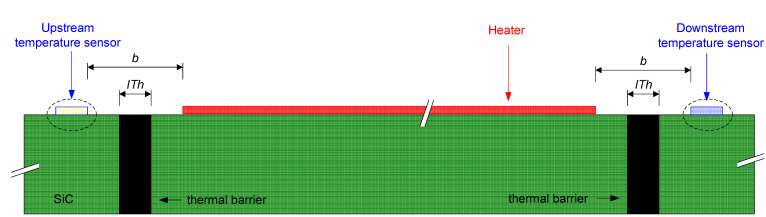
TMFS geometry with thermal barriers for COMSOL Multiphysics simulation, showing the temperature sensors and heater. Figure is not to scale.

**Figure 6 sensors-22-03484-f006:**
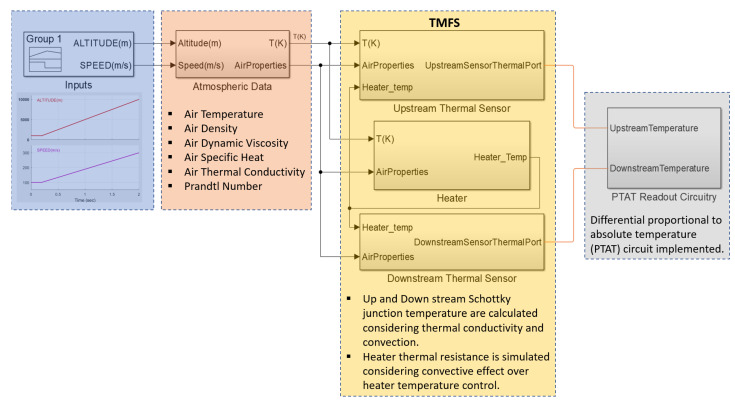
Simulink model block diagram.

**Figure 7 sensors-22-03484-f007:**
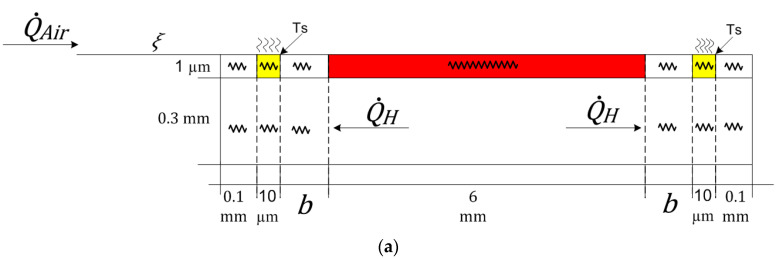
Side view of idealized TMFS layout with thermal resistances and heat flux for (**a**) without and (**b**) with thermal barrier.

**Figure 8 sensors-22-03484-f008:**
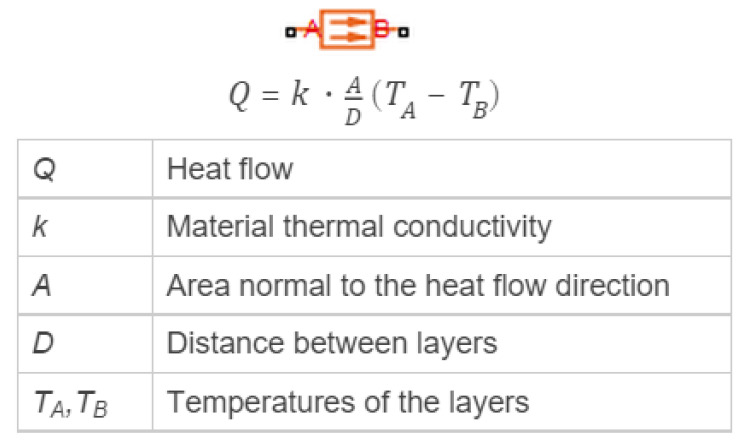
Conductive heat Simulink Simscape block implementation by Mathworks.

**Figure 9 sensors-22-03484-f009:**
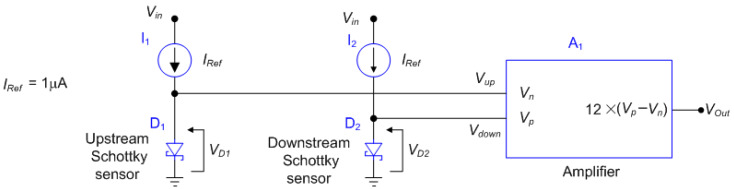
TMFS differential PTAT circuit.

**Figure 10 sensors-22-03484-f010:**
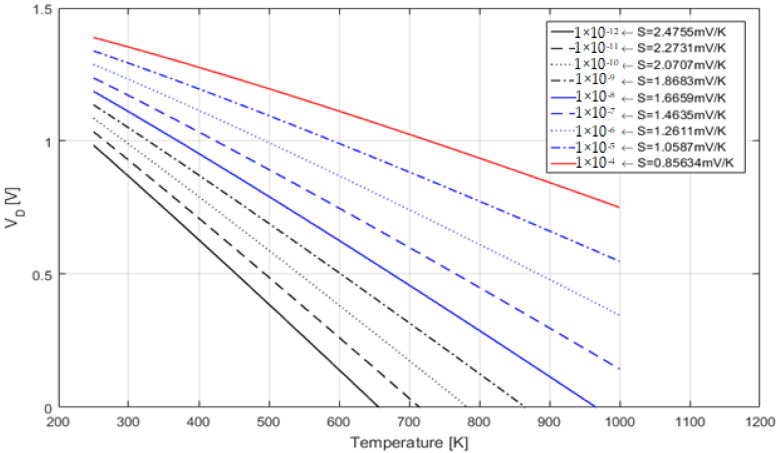
Schottky diode Simulink model validation: I_Ref_ in amps and the correspondent Schottky diode sensitivity in mV/K.

**Figure 11 sensors-22-03484-f011:**
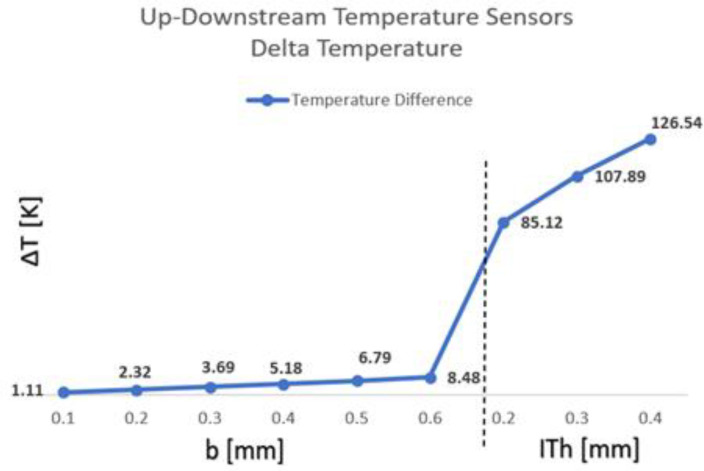
Calorimetric TMFS upstream and downstream temperature difference for each assessed condition, where the dashed line separates the increase in “b” and the introduction and variation in “IT_h_” thickness for a fixed “b” of 0.6 mm—COMSOL Multiphysics.

**Figure 12 sensors-22-03484-f012:**
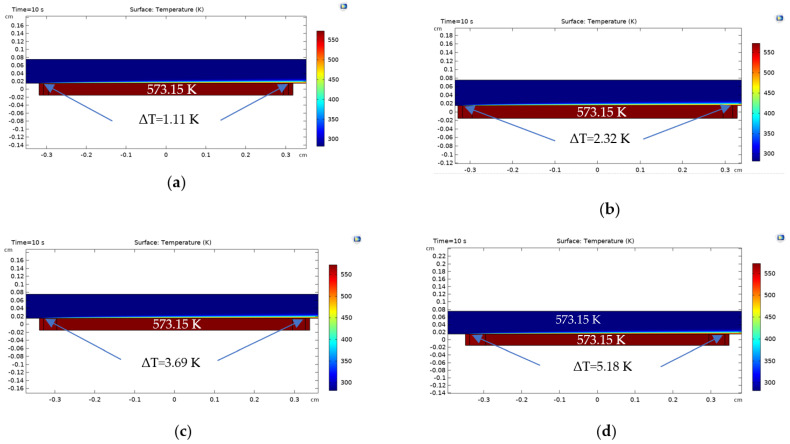
CFD results: (**a**) b=0.1 mm, (**b**) b=0.2 mm, (**c**) b=0.3 mm, (**d**) b=0.4 mm, (**e**) b=0.5 mm, (**f**) 0.6 mm, (**g**) $b=0.6\;\mathrm{mm}\;|\;IT_{h}=0.2\;\mathrm{mm}$, (**h**) $b=0.6\;\mathrm{mm}\;|\;IT_{h}=0.3\;\mathrm{mm}$, and (**i**) $b=0.6\;\mathrm{mm}\;|\;IT_{h}=0.4\;\mathrm{mm}$ —COMSOL Multiphysics.

**Figure 13 sensors-22-03484-f013:**
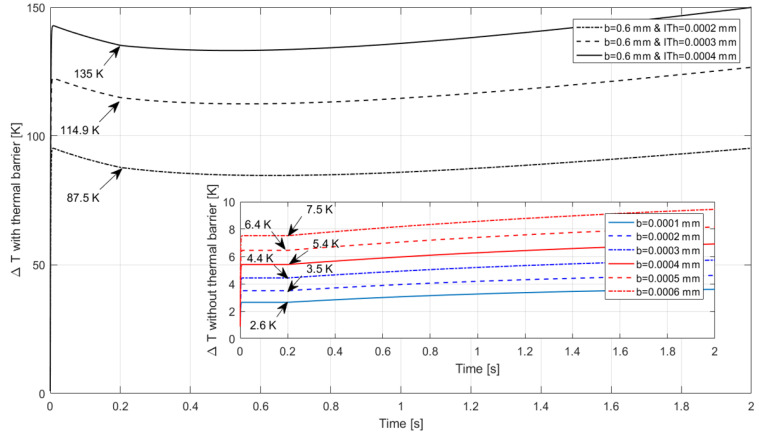
Measured ΔT between upstream and downstream sensors. The plot in the background presents the ΔT results with the thermal barrier and the plot in the foreground presents the ΔT results without the thermal barrier. The arrows point to the temperature difference measured at 0.2 s of the simulation—Simulink.

**Figure 14 sensors-22-03484-f014:**
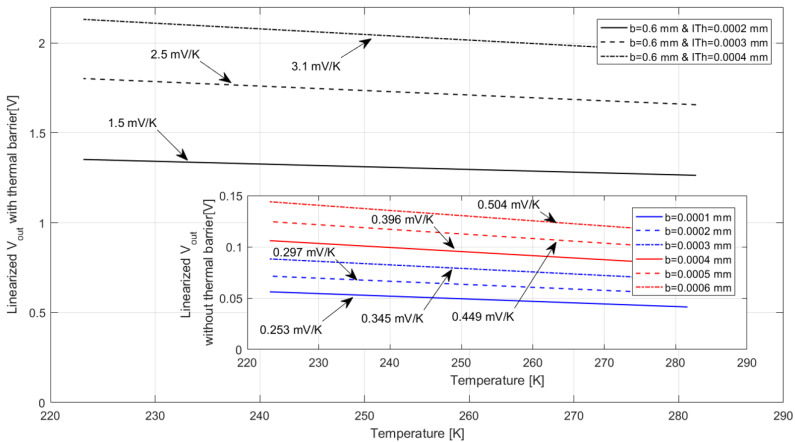
Linearized V_out_ from PTAT circuit as a function of air temperature and respective sensitivities to temperature variation. The plot in the background presents the TMFS V_out_ and sensitivity results with the thermal barrier and the plot in the foreground presents the TMFS V_out_ and sensitivity results without the thermal barrier. The arrows point to the sensitivity to air temperature variation of each TMFS design assessed during the simulation—Simulink.

**Figure 15 sensors-22-03484-f015:**
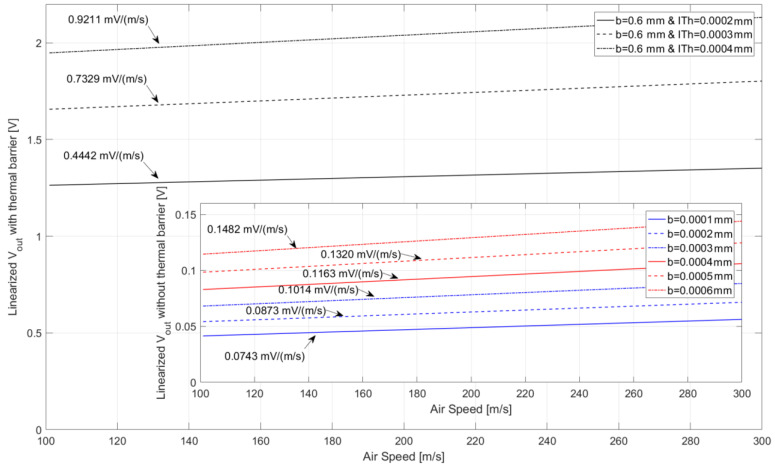
Linearized V_out_ from the PTAT circuit as a function of air speed and respective sensitivities to speed variation. The plot in the background presents the TMFS V_out_ and sensitivity results with the thermal barrier and the plot in the foreground presents the TMFS V_out_ and sensitivity results without the thermal barrier. The arrows point to the sensitivity to air speed variation of each TMFS design assessed during simulation—Simulink.

**Table 1 sensors-22-03484-t001:** Material properties.

Property	Unit	Substrate (SiC)	Insulation (Parylene N)	Heater (W)	Thermal Sensor (Au)
**Electrical Resistivity**	Ωm @ 20 °C	10	-	5.65 × 10^−8^	2.44 × 10^−8^
**Thermal Conductivity**	W/m—K	114	0.126	174	300
**Young’s Module**	Nmm^2^	4.15 × 10^5^	-	4.11 × 10^5^	79 × 10^3^
**Poisson’s Ratio**	N/A	0.16	-	0.280	0.42
**Density**	gcm^−3^ @ 25 °C	3.16	1.11	19.3	19.32
**Melting Point**	°C	2830	-	3687	1064
**Specific Heat Capacity**	J/g-°C	0.670	1.3	0.134	0.133

**Table 2 sensors-22-03484-t002:** Air properties as per ISA standard.

Property	Unit	Value
**Dynamic Viscosity**	μPa.s	17.606
**Sound Speed**	m/s	336.766
**Kinematic Viscosity**	μm^2^/s	15.705
**Thermal Conductivity**	W/m·K	0.024874
**Altitude**	m	1000
**Temperature**	K	282.207
**Pressure**	hPa	90,813

**Table 3 sensors-22-03484-t003:** Mesh node numbers and quality.

$b/IT_{h}\;\mathrm{in}\;\mathrm{mm}$	0.1	0.2	0.3	0.4	0.5	0.6	0.6/0.2	0.6/0.3	0.6/0.4
**Domain**	1,427,329	1,464,080	1,500,454	1,535,496	1,571,820	1,607,178	1,602,022	1,602,876	1,601,974
**Boundary**	29,378	29,967	30,549	31,126	31,698	32,266	32,520	32,526	32,540
**Minimum Quality**	0.4875	0.4485	0.4491	0.4386	0.4748	0.4484	0.4484	0.4484	0.4484

**Table 4 sensors-22-03484-t004:** Comparison of COMSOL Multiphysics and Simulink ΔT results.

B in mm	${\mathrm{IT}}_{h}\mathrm{in}\;\mathrm{mm}$	ΔTfrom COMSOL Multiphysics in Kelvin	ΔTfrom Simulink in Kelvin	Error in Kelvin (%)
0.1	0	1.1	2.6	1.5 (58%)
0.2	0	2.32	3.5	1.18 (34%)
0.3	0	3.69	4.4	0.71 (16%)
0.4	0	5.18	5.4	0.22 (4%)
0.5	0	6.79	6.4	0.39 (6%)
0.6	0	8.48	7.5	0.98 (13%)
0.6	0.1	85.12	87.5	2.38 (3%)
0.6	0.2	107.89	114.9	7.01 (6%)
0.6	0.3	126.54	135	8.46 (6%)

## Data Availability

Models created for simulations herein presented can be shared upon request to authors as per the contact information provided above.

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
