# Peer review of "High-Speed and High-Temperature Calorimetric Solid-State Thermal Mass Flow Sensor for Aerospace Application: A Sensitivity Analysis"

_sensors, 2022, doi:10.3390/s22093484_

Round 1

Reviewer 1 Report

The authors manage to improve their work significantly comparing to the previous submission, but they're still some issues to be addressed.

  1. As I mentioned in my previous review “the most important remark is that this work is totally based on simulation results. The main model was based on a 2D (!) Comsol geometry combined with a corresponding Simulink model. There are already many more sophisticated 3D models, which were evaluated in detail and experimentally verified. I cannot see any innovation in the present approach which is extremely simplified in a field where several mature devices were already reported and evaluated.”
    Unfortunately, the entire work is still based on a “theoretical” device that is simulated by a 2D Comsol model. I’m sorry to say that the authors’ claim that “3D models and simulations tools may give better results, but they demand computational resources, which are not always available to the academia” is not true. For this simple structure, a 3D model FEA simulation can be easily performed by a common PC (eg i7, 16-32GB RAM) and does not require expensive resources.
  1. Furthermore, the claim that “This approach is highly used in CFD simulations once it reduces the size of the meshes generated, saving computation resources, without compromising the outcome results” is also not correct. It is not a matter of mesh size, but it involves different geometry approaches. For example, from the top view of the device, which is presented in fig. 2 there is a lateral thermal conductive path between the heater and the temperature sensor, through the substrate. This thermal conductive path disappears in fig. 5, which represents the 2D model, so the heater and the temperature sensors communicate thermally only through the thermal barrier. This is only due to the fact that the 2D model cannot accurately simulate the real structure, which induces significant deviations from the obtained results. I would recommend to the authors to use the top-view indicated in fig. 2 in order to perform the 2D simulations, so as to find the differences with the 2D cross-sectional geometry that they’ve used. In a 3D model, the temperature distribution should be exactly the same in both cases.
  2. And this brings us to the third major issue, which is the device fabrication in the real world. In this version a phrase was added in the manuscript: “The insulator has been chosen due to its low thermal conductivity and compatibility with microfabrication process, which filling into the insulation trench in the SiC substrate can be achieved by Chemical Vapor Deposition (CVD) process having the excess further etched using Reactive Ion Etching (RIE) process via oxygen gas [29,30].” However, ref 29 demonstrates the deposition of Parylene film on a SiC substrate using CVD without any patterning. The SiC substrate was flat without any topology or trenches. Ref. 30 involves the use of parylene in post CMOS packaging, which is not related to the present geometry. According to fig. 2 and 5, a high anisotropic through-wafer etching should be performed. The lateral length should be 3mm, while the minimum width should be equal to ITh thickness, which is 0.1mm. The authors should describe in detail the proposed possess steps (including substrate patterning, Paylene deposition and patterning) in order to fabricate such a device.
  1. The boundary conditions for the Comsol simulation should be clearly defined. The mode of operation should be also indicated (constant power, or constant temperature). According to the literature, an optimum temperature sensor distance should exist in terms of sensitivity, thus simulations should be extended beyond B=0.6mm in order to find this distance. Normally there should be a value for B that above it, the sensitivity should be decreased.
  2. The term ‘High Speed’ in the title is not supported with the corresponding results in the manuscript, as there is no dynamic evaluation of the device, therefore, it should be omitted.

Author Response

The authors manage to improve their work significantly comparing to the previous submission, but they're still some issues to be addressed.

  1. As I mentioned in my previous review “the most important remark is that this work is totally based on simulation results. The main model was based on a 2D (!) Comsol geometry combined with a corresponding Simulink model. There are already many more sophisticated 3D models, which were evaluated in detail and experimentally verified. I cannot see any innovation in the present approach which is extremely simplified in a field where several mature devices were already reported and evaluated.”
    Unfortunately, the entire work is still based on a “theoretical” device that is simulated by a 2D Comsol model. I’m sorry to say that the authors’ claim that “3D models and simulations tools may give better results, but they demand computational resources, which are not always available to the academia” is not true. For this simple structure, a 3D model FEA simulation can be easily performed by a common PC (eg i7, 16-32GB RAM) and does not require expensive resources.

Regarding the reviewer's comment, would like to point to the article “Why is Meshing Important for Structural FEA and Fluid CFD Simulations?” in the Simutech group, a working group operating in partnership with Ansys, which state themselves as:

“…the largest Ansys Elite Channel Partner in North America, we support and empower engineers all over the globe with our engineering expertise in simulation software and value-added services”

Below, follows the link to the article:

Why is Meshing Important for Structural FEA and Fluid CFD Simulations? (simutechgroup.com)

Ansys is one of the largest FEA and CFD simulation tools and most consolidated in the market being widely used by the Academia and Industry.

From the above article, is possible to understand the relationship between the mesh granularity and the simulation convergence criteria. Table 3 in the submitted paper presents the quality data about the mesh generated for the COMSOL simulation as well as the section 4.1 presents the criteria used to generate the mesh. The minimum element size was equal to 0.1um, 10 times lower than the lowest Calorimetric TMFS dimension, which is 1um for the metals (gold and tungsten) thickness. This criterion has allowed the software to achieve the convergence criteria for all simulated conditions.

In the article on the above link, it is also possible to check that FEA (Finite Element Analysis) is only used for structural simulations not being recommended for CFD simulations and the rationales for that are also provided in the text.

Therefore, the meshing criteria which has generated approximately 1,5 Million nodes for each evaluated case would never be simulated as a 3D geometry in a common PC (eg i7, 16-32GB RAM). Considering the calorimetric TMFS lateral length of 3mm and the minimum node element size of 0.1um, the 1,5 Million node counting should be multiplied by 10,000 giving approximately 15 billion nodes to resolve, beyond the capacity of any common desk top specification.

  1. Furthermore, the claim that “This approach is highly used in CFD simulations once it reduces the size of the meshes generated, saving computation resources, without compromising the outcome results” is also not correct. It is not a matter of mesh size, but it involves different geometry approaches. For example, from the top view of the device, which is presented in fig. 2 there is a lateral thermal conductive path between the heater and the temperature sensor, through the substrate. This thermal conductive path disappears in fig. 5, which represents the 2D model, so the heater and the temperature sensors communicate thermally only through the thermal barrier. This is only due to the fact that the 2D model cannot accurately simulate the real structure, which induces significant deviations from the obtained results. I would recommend to the authors to use the top-view indicated in fig. 2 in order to perform the 2D simulations, so as to find the differences with the 2D cross-sectional geometry that they’ve used. In a 3D model, the temperature distribution should be exactly the same in both cases.

Concerning the model meshing, we clarify our point in 1).. Regarding the 2D geometry and the heat path mentioned, we want to bring the below picture for the discussion:

If the heat path represented by the orange arrow is the one you are mentioning, your point is correct with regards to the fact the 2D model do not capture it once it is volumetric dependent. However, it is important to highlight that the cross-section area free for this heat path to form is very low which will limit the amount of heat passing through it representing a limited influence in the presented results.

Regarding a top-view 2D simulation, it will not bring any value to the results being reported once it is not possible to capture the convective effects from the air flowing on top of the sensors, otherwise, it would be a 3D simulation.

Please, refer to reference 21 in the paper for similar approach used in the literature.

  1. And this brings us to the third major issue, which is the device fabrication in the real world. In this version a phrase was added in the manuscript: “The insulator has been chosen due to its low thermal conductivity and compatibility with microfabrication process, which filling into the insulation trench in the SiC substrate can be achieved by Chemical Vapor Deposition (CVD) process having the excess further etched using Reactive Ion Etching (RIE) process via oxygen gas [29,30].” However, ref 29 demonstrates the deposition of Parylene film on a SiC substrate using CVD without any patterning. The SiC substrate was flat without any topology or trenches. Ref. 30 involves the use of parylene in post CMOS packaging, which is not related to the present geometry. According to fig. 2 and 5, a high anisotropic through-wafer etching should be performed. The lateral length should be 3mm, while the minimum width should be equal to ITh thickness, which is 0.1mm. The authors should describe in detail the proposed possess steps (including substrate patterning, Paylene deposition and patterning) in order to fabricate such a device.

We do not believe that a specific fabrication process needs to be determined in the paper once it is outside the paper scope, which presents the results of a sensitivity assessment performed where the methodology adopted is exemplified.

  1. The boundary conditions for the Comsol simulation should be clearly defined. The mode of operation should be also indicated (constant power, or constant temperature). According to the literature, an optimum temperature sensor distance should exist in terms of sensitivity, thus simulations should be extended beyond B=0.6mm in order to find this distance. Normally there should be a value for B that above it, the sensitivity should be decreased.

The mechanical boundary conditions for the calorimetric TMFS proposed are presented on table 1 and the airflow boundary conditions are presented on table 2. In the submitted paper text, it is often mentioned that the sensor temperature is kept constant at 573K (300ºC), which is the definition for a calorimetric sensor (constant temperature operation).

  1. The term ‘High Speed’ in the title is not supported with the corresponding results in the manuscript, as there is no dynamic evaluation of the device, therefore, it should be omitted.

To obtain the sensitivity information there shall have variation in time and this is obtained from the Simulink model and represented in the Figure 14 showing the sensor sensitivity to airflow speed variation up to 300m/s. This is also stated on section 4.2.

Reviewer 2 Report

The authors have adequately dealt with all previous remarks.

Author Response

Thanks

Round 2

Reviewer 1 Report

The manuscript remains the same, and the authors fail to address my main remarks, namely:

  1. Adaptive meshing techniques can be used to minimize nodes and still get useful results with 3D models. In the current approach, there are approximately 1,5 Million nodes and the authors claim that: “Considering the calorimetric TMFS lateral length of 3mm and the minimum node element size of 0.1um, the 1,5 Million node counting should be multiplied by 10,000 giving approximately 15 billion nodes to resolve”. This is not the case, since the structure is very simple and node element size does not need to be 0.1um on the y-axis. In fact, an average size (considering adoptive messing) of 1-2um is more than enough since the elements need to be small only at the end of each structure line. Additionally, given the symmetry of the geometry, only half of the device needs to be meshed, thus the final nodes are well below 15B and can be processed by a common high-end system.

Many groups have already published 3d simulations in more complex REAL devices (eg.

 https://www.comsol.com/blogs/describing-the-behavior-of-a-thermal-mass-flow-sensor/

https://www.mdpi.com/1424-8220/22/3/1056 )

  1. The authors accept that “your point is correct with regards to the fact the 2D model do not capture it once it is volumetric dependent” but they conclude that “the cross-section area free for this heat path to form is very low which will limit the amount of heat passing through it representing a limited influence in the presented results”. How do they know that the specific geometrical characteristic (which they do not take into consideration in the simulation), has limited influence on the results? The thermal conductivity of SiC is high, thus a significant amount of heat will be transferred to the temperature sensor through this path.

The authors also indicate ref. 21 for a similar simulation approach. Ref. 21 presents a thermal mass flow sensor that was actually fabricated using silicon bulk micromachining. This REAL device was also characterized in laboratory conditions and the simulation was an additional tool to help in the device geometry optimization. This cannot be compared to the present manuscript, where a theoretical device is simulated and this simulation is actually the core subject of the work.

In fact, I cannot recall any work in any prestigious and high-quality scientific journal that refers only to 2D simulation of a theoretical device.

  1. The authors believe that they do not need to present a specific fabrication process because it is outside of the paper's scope. However, they have to prove that this “theoretical device” that they use in the simulation can be fabricated by standard techniques otherwise the simulation they present is meaningless.

In conclusion, I cannot propose the publication of this manuscript unless (at least) the following additions are made:

  1. 3D device simulation and comparison of the corresponding results with the current approach.
  2. Presentation of standard process flow steps for the fabrication of the proposed device.

Round 3

Reviewer 1 Report

In this review round, the authors presented a descent 3D simulation approach, where they describe in detail the process they follow.

I have to make the following remarks:

The resources they use can marginally support the project they have to accomplish. To be more specific, if a research group plan to work in the field of simulation (especially using COMSOL, ANSYS etc) then a mid-range laptop with 16GB RAM is not a way to proceed. I can understand the resource shortage and I respect the effort, but under these circumstances is difficult to produce novel and accurate results.

The authors in the first part of their simulations are using a meander-shaped heater, which is not mentioned in the manuscript. When I mentioned ‘simple structure’ I was referring to an orthogonal heater structure which is presented in fig. 2 of the manuscript.

According to the report the authors manage to perform 3D simulations in the case of square-shaped heater, but the messing was coarse. All the messing issues were caused by the low RAM of the laptop, which as I mentioned above is not ideal equipment for such processes. I guess if they use a PC with more RAM (32GB or more), they will manage to perform detailed and accurate 3D simulations.

Even with this coarse 3D approach, useful results can be extracted. The results are presented in fig.1 of the report which is compared to fig. 11 of the manuscript.

It is obvious that in the 3D results the temperature differences with and without the insulation barrier are significantly lower. Specifically, the difference in the 2D case is about 10 times higher, while in the 3D case is only 4 times higher. This is probably due to the thermal path that I mentioned to my previous review. Furthermore, the increasing effect of the ITh thickness is totally different. In the 2D case, there is an almost linear increase relation, while in the 3D case a saturation is observed.

Regarding the fabrication process, I can accept the general concept even if there are steps that cannot be performed in a real lab. For example, after the CVD deposition and the corresponding lift-off of the thick (300um) parylene layer the developed topography will not allow further process without a planarization step. In any case, it should be clear that this is a proposed process; thus, the corresponding text should be modified as follow:

“The calorimetric TMFS thermal barrier can be fabricated by making use of standard cleanroom microfabrication techniques. The proposed process is only a coarse description of an actual fabrication process flow that is, in practice, much more complicated. This process can start with a commercial SiC wafer (300 µm thick). Photoresist is spin-coated on the wafer, following a UV-exposure with a photomask with the thermal barrier area design. The area for SiC etching is defined by the photolithography process, and a trench can be etched on SiC via Inductive Coupled Plasma RIE, as described in [47]. Following, a new photolithography process is conducted to define the areas for Parylene-N deposition, which can then can be performed via CVD. The photoresist is then removed in an Acetone bath, and the SiC wafer containing the Parylene-N trenches is cleaned via RCA cleaning.

The Schottky diodes electrodes and heater can be defined by a subsequent photolithography process, followed by tungsten and gold deposition via e-beam metal deposition. Resist removal and lift-off are performed with an Acetone bath. The final step will be the sensors dicing with a diamond dicing saw and wiring-bonding to a PCB.”

In conclusion, I appreciate the effort to support their work and, even if I have all the concerns mentioned above, I will recommend publication at this stage. However, I strongly suggest improving this work by getting access to a high-level PC in order to obtain more accurate results.

Author Response

Dear Reviewer, dear Editor,

As per comments received from reviewer 1 on Round 2 the submitted paper was updated including the following information:

  • 3D CFD Simulations: Section 4.1 “COMSOL Multiphysics” was updated including the following statement as result of the assessment performed in our reply to Reviewer’s 2nd round:
    1. “For comparison a 2D and 3D simulation was conducted and showed that the y-direction temperature dependence into the calorimetric TMFS body temperature distribution was limited to a small magnitude due to low cross-section area available for the thermal conductive path to form in a way that it can not significantly impact the results obtained from a 2D simulation. Therefore, each proposed design...”
  • Section 4.3 “Calorimetric TMFS Fabrication Process” was included and updated as per reviewer’s request as:

“The calorimetric TMFS thermal barrier can be fabricated by making use of standard cleanroom microfabrication techniques. The proposed process is only a coarse description of an actual fabrication process flow that is, in practice, much more complicated. This process can start with a commercial SiC wafer (300 µm thick). Photoresist is spin-coated on the wafer, following a UV-exposure with a photomask with the thermal barrier area design. The area for SiC etching is defined by the photolithography process, and a trench can be etched on SiC via Inductive Coupled Plasma RIE, as described in [47]. Following, a new photolithography process is conducted to define the areas for Parylene-N deposition, which can then can be performed via CVD. The photoresist is then removed in an Acetone bath, and the SiC wafer containing the Parylene-N trenches is cleaned via RCA cleaning.

The Schottky diodes electrodes and heater can be defined by a subsequent photolithography process, followed by tungsten and gold deposition via e-beam metal deposition. Resist removal and lift-off are performed with an Acetone bath. The final step will be the sensors dicing with a diamond dicing saw and wiring-bonding to a PCB.”

  • Minor spellings were correct in the whole paper;
  • Number separation character on Table 4 was replaced from “,” to “.”

The revision marks are enabled in the version submitted as per request.

We would like to take the opportunity and thank the reviewer 1 and the Editor for all support and attention dedicated to us in the submission and review process.

Best Regards